# Searchable Blockchain-Based Healthcare Information Exchange System to Enhance Privacy Preserving and Data Usability

**DOI:** 10.3390/s24051582

**Published:** 2024-02-29

**Authors:** Sejong Lee, Yushin Kim, Sunghyun Cho

**Affiliations:** 1Department of Computer Science and Engineering, Major in Bio-Artificial Intelligence, Hanyang University, Ansan 15588, Republic of Korea; kingsaejong@hanyang.ac.kr (S.L.); hpwgg045@hanyang.ac.kr (Y.K.); 2Department of Computer Science and Engineering, Hanyang University ERICA, Ansan 15588, Republic of Korea

**Keywords:** blockchain, healthcare information exchange, privacy preserving, data usability

## Abstract

Ensuring the security and usability of electronic health records (EHRs) is important in health information exchange (HIE) systems that handle healthcare records. This study addressed the need to balance privacy preserving and data usability in blockchain-based HIE systems. We propose a searchable blockchain-based HIE system that enhances privacy preserving while improving data usability. The proposed methodology includes users collecting healthcare information (HI) from various Internet of Medical Things (IoMT) devices and compiling this information into EHR blocks for sharing on a blockchain network. This approach allows participants to search and utilize specific health data within the blockchain effectively. The results demonstrate that the proposed system mitigates the issues of traditional HIE systems by providing secure and user-friendly access to EHRs. The proposed searchable blockchain-based HIE system resolves the trade-off dilemma in HIE by achieving a balance between security and the data usability of EHRs.

## 1. Introduction

Healthcare information exchange (HIE) systems face a critical dilemma in balancing robust privacy preserving with the usability of electronic health records (EHR). In digital format, EHRs encompass a patient’s healthcare information, including vital signs, medication records, allergies, vaccinations, sleep patterns, and personal details [1,2]. With the commercialization of EHRs, the demand for remote medical services and medical artificial intelligence (AI) models using EHRs has increased, fueling the growth of advanced medical industries [3,4]. However, owing to privacy issues and closed management systems in healthcare, there is a shortage of available EHRs [5,6,7]. HIE systems enabling healthcare information sharing through networks have emerged to address this issue [8,9]. Nowadays, HIE extends beyond information exchange between healthcare providers, encompassing technologies allowing individuals to manage their health data collected through the Internet of Medical Things (IoMT) and share them with other medical providers or relevant third parties [10,11].

Recent research on HIE systems is focused on applying distributed storage technologies such as cloud and blockchain into systems to consider realistic EHR management scenarios [12,13,14]. Blockchain is garnering significant attention for its ability to ensure data integrity and reliability. In blockchain-based HIE systems, data stored in the ledger cannot be altered or deleted without the consensus of a majority of participants. This mechanism offers users enhanced reliability and integrity of EHRs. Unlike cloud-based HIE systems, blockchain-based systems operate without a central administrator, reducing the possibility of third-party intervention in exchanging EHRs. This characteristic significantly reduces the risk of privacy breaches and facilitates the design of safer HIE systems. Therefore, many researchers are investigating blockchain-based HIE systems [15].

In blockchain-based HIE systems, users’ privacy and healthcare information in EHRs is securely shared through encryption [16]. The encryption for EHRs often employs key-based technology, such as symmetric or public-key cryptography. In symmetric-key cryptography, the same key is used for encryption and decryption, necessitating key sharing when exchanging data. Public-key cryptography uses different keys for encryption and decryption, requiring the secret key, which corresponds to the public key, for decryption. Although key-based encryption technology offers robust security for EHRs, it also presents challenges regarding key management and data usability. In key-based encryption technology, the encryption key that can decrypt the data signifies ownership of the data. Control can be lost if the key is compromised during decryption sharing. Therefore, protecting EHRs with key-based encryption requires using either the data requester’s public key for each encryption or one-time keys. This approach leads to storage issues in blockchain-based HIE systems because encrypted data is saved in duplicate. Furthermore, since the contents of the encrypted EHR are inaccessible until decrypted, the utility of the data is diminished.

Research is being conducted to apply flexible encryption technologies, such as searchable and homomorphic encryption, into blockchain-based HIE systems to balance privacy and data usability [17,18,19,20,21]. Searchable encryption is a cryptographic technique that allows information to be searched for using specific keywords while maintaining confidentiality, even in its encrypted state. Han et al. proposed a system that combines attribute-based access control (ABAC) and blockchain-based searchable encryption [19]. Implementing attribute-based access control and searchable encryption processes through smart contracts provides greater granular access control and flexible search capabilities. Their approach mitigated the trade-off issue in blockchain-based data-sharing environments. Homomorphic encryption can calculate encrypted data without decryption, offering high usability while preserving privacy. Ali et al. proposed HealthLock, which combines homomorphic encryption with blockchain technology to secure and efficiently share healthcare information in medical IoT environments [21]. The nature of homomorphic encryption enables operations on encrypted data, thus protecting user privacy throughout the data-sharing process. In addition, automated access control and policy management through smart contracts enhance security while improving data usability. These studies improved the data usability of HIE by introducing flexible encryption techniques capable of operating on encrypted data and access control mechanisms. However, limitations still exist, such as the need to share keywords for searching encrypted data or keys for decryption, which do not fully resolve the trade-off issue. The primary goal of HIE is to share EHRs securely and utilize them efficiently. Therefore, research on balanced HIE systems that provide strong security while enhancing data usability is necessary [22].

This study proposes a searchable blockchain-based HIE system that resolves the trade-off issue by preserving privacy while providing high data usability. In the proposed system, users generate healthcare information collected from IoT devices into an EHR block and share it with other users through the blockchain. Participants in a blockchain can use a hash of desired information to search for specific health data within the blockchain. Users obtained through attribute-based search schemes can directly utilize health information without needing separate decryption or data sharing. Unlike conventional HIE systems that focus on processing EHRs, our proposed system handles the information in EHRs. By sharing detailed information in EHRs, our system enhances data usability in the HIE system while maintaining robust security, thereby resolving the trade-off problem. The primary contributions of this study are as follows.

We present the trade-off issue in HIE systems, where the robust security measures implemented for privacy preserving diminished data usability.We propose a blockchain-based HIE system to solve the trade-off issue in HIE, allowing for the searching of encrypted EHR information.We present direct examples of utilizing healthcare information stored in a blockchain-based HIE system.

The remainder of this paper is organized as follows. Section 2 discusses the research related to blockchain-based HIE systems. Section 3 defines the problem using the proposed system model. Section 4 provides a detailed explanation of the proposed searchable blockchain-based HIE system. In Section 5, we evaluate the performance of the proposed system and compare it with that of other related systems. Finally, Section 6 concludes the study.

## 2. Related Work

This section introduces the research cases of blockchain-based HIE systems that apply security techniques with utility considerations to address the trade-off between privacy preserving and data usability.

Chen et al. proposed a blockchain-based searchable encryption scheme to protect personal information while efficiently sharing the EHRs [17]. Their approach uses complex boolean expressions stored in a blockchain, creating multiple indices for EHR information. EHRs are encrypted using a symmetric key and stored separately on a cloud server. Users in this system retrieve EHR information via smart contracts, using indices as inputs to access actual EHRs. This approach, which employs multiple indices for EHRs, has the potential to reduce privacy exposure and control access to EHRs. However, this does not entirely resolve the trade-off between privacy performance and data usability in blockchain-based HIE systems. Creating indices using Boolean expressions provides anonymity for range-based items, such as age or specific numerical values. However, when the variety of possible information types is limited, exposure to an item could facilitate inferences about the actual EHR, posing a risk. This risk is heightened by dictionary- or graph-based re-identification attacks, which exploit limited information ranges to enhance attack success rates [23,24,25]. Furthermore, using symmetric key encryption for EHRs raises data ownership issues, as sharing the key equates to sharing the data ownership.

Niu et al. proposed a blockchain for HIE that resolves the trade-off problem using an attribute-based searchable encryption technique [18]. They proposed a private blockchain-based HIE platform comprising medical institutions that manage and store encrypted local EHRs. Hospitals in this network shared specific EHR keywords as transactions. This system categorized user nodes as either general patients, the subjects of the EHRs, or doctors, the creators of the EHRs. Access to EHRs is controlled by varying the connected keywords using polynomials tailored to user attributes. The authors sought to resolve this trade-off by offering refined access control and precise data retrieval through a private blockchain with attribute-based encryption and polynomial-based keyword generation. However, administrators’ existence in proposed blockchain-based systems introduces ambiguity to decentralization. Moreover, the lack of a mechanism to provide actual EHR information has incompletely resolved the trade-off issue.

Li et al. proposed EHRChain, a blockchain-based HIE system incorporating attribute and homomorphic encryption techniques to address the trade-off problem [20]. They also introduced a cryptographic primitive named semi-policy hiding and dynamic permission changing for partial ciphertext-policy attribute-based encryption (SHDPCPC-CP-ABE). This primitive effectively controls access to EHRs by employing a semi-policy hiding approach, allowing access only to users with authorized attributes, and a dynamic permission-changing technology for modifying the access policies of EHRs stored encrypted on the blockchain. In EHRChain, the Improvement Paillier encryption technology prevents privacy exposure using patient information during insurance processing [26]. Insurance companies and medical practitioners encrypt the disease information of their EHRs using the improved Paillier technology. This technique compares encrypted areas to ascertain the authenticity of the information. Using improved Paillier encryption, which reveals only the disease-related area of the EHR, the system minimizes privacy exposure threats that could arise during the utilization of EHRs. EHRChain, proposed by the authors, mitigates the trade-off problem in HIE by offering flexible access control and data protection technology. However, the system faces trust issues when updating the hash indices of EHRs stored in IPFS during policy changes for flexible access control. A solution is required to ensure the integrity of the updated hash indices and confirm that they correspond to the requested EHRs. Additionally, the Paillier encryption technology, while safeguarding data privacy, suffers from processing delays owing to the complex calculations inherent in homomorphic encryption and limited usability due to restricted information usability.

Various studies attempting to resolve the trade-off problem in blockchain-based HIE systems have proposed improved access control or an indirect approach to provide information, thereby enhancing data usability [21,27]. However, using EHRs requires sharing the decryption key or relying on limited information. Although privacy security is crucial in systems handling EHRs, HIE’s goal to share and utilize a vast amount of healthcare information is equally vital. Therefore, research into blockchain-based HIE systems that balance privacy protection with data usability is necessary.

## 3. Problem Definition

### 3.1. System Model

The proposed system comprises four key components that collaborate to exchange healthcare information within the network. Figure 1 shows an overview of the proposed system. All IoMT devices periodically check a user’s health status and generate health-related information. The EHR owner compiles this information from IoMT devices to create and manage their EHR. The healthcare information included in EHR is stored in the HIE blockchain as a block. An EHR searcher uses a blockchain to search for and utilize healthcare information by employing records or hash values.

#### 3.1.1. Searchable Blockchain

The searchable blockchain is a distributed ledger shared among all the users of the HIE system. Figure 2 shows the structure of the EHR block used in the proposed system. The EHR block in our blockchain consists of a block header containing information about the block and a block body where transactions related to healthcare information are stored. The header of an EHR block includes the hash value of the previous block, Merkle root, and the block’s creation time, similar to the headers in typical blockchain blocks. However, it differs because it does not include a nonce or bits for mining difficulty used to regulate block creation time. In the proposed blockchain, the EHR owner, who is the subject of the information, creates a block using the EHR and no mining process is involved. The body of an EHR block contains healthcare information extracted from the EHR and is formatted as a transaction. Because each block body consists of information extracted from a single EHR, each block represents one EHR.

#### 3.1.2. Internet of Medical Things (IoMT)

IoMT devices continuously monitor users’ health status and generate personal healthcare information. Health data generated by IoMT devices are transmitted to the user’s smart device or computer. The IoMT includes wearable devices, applications, implants, and smart home sensors for diagnosis, treatment, and health management. Each IoMT device was assumed to produce different types of healthcare information to prevent redundancies in the blockchain.

#### 3.1.3. EHR Owner

The EHR owner is a crucial participant in our proposed system and is responsible for generating and sharing healthcare information. EHR owners range from individuals with chronic conditions requiring continuous health management to general users and medical professionals. They use the healthcare information collected from their IoMT to create and manage their EHRs. EHR owners may share their EHRs commercially or as part of medical research through the blockchain. They anonymize their healthcare information using hashing algorithms such as SHA-256 and create healthcare information transactions to share EHRs. Because EHRs contain more than healthcare information, indiscriminate sharing can expose privacy. Therefore, unique, identifiable information was minimized by utilizing quasi-identifiers and healthcare information to preserve privacy. The transactions form the EHR blocks and the user propagates these blocks to the blockchain network to record the data in the ledger. EHR owners participate in the blockchain network as lightweight nodes to reduce storage load, carrying a reduced version of the blockchain information rather than full nodes.

#### 3.1.4. EHR Searcher

Users participating in a blockchain network may have different roles depending on the situation. A node that plays the role of an EHR owner sharing its own EHR information can become an EHR searcher when necessary. The EHR searcher plays the role of a consumer in the proposed system by utilizing the EHR stored in the blockchain. EHR searchers include medical professionals, researchers, public health authorities, and individual users who use healthcare information to track infectious diseases, medical AI research, treatment, and health management. EHR searchers utilize the search functionality to locate desired EHRs or specific healthcare information within the blockchain. The users can employ various search parameters like records, hash values, or specific attributes for efficient EHR searching. EHR searchers interested in acquiring diverse health information can store all block data of the blockchain network, acting as full nodes.

### 3.2. Threat Model

We consider various security threats that pose privacy concerns within our blockchain-based HIE system. The security threats under consideration occur within a probabilistic polynomial-time framework. Given that EHRs shared through a public blockchain are accessible to anyone within the network, security attacks aimed at breaching privacy can be perpetrated by malicious users or honest but curious participants within the blockchain network. The security attacks considered in our proposed HIE system include the following:Inference Attacks: Inference attacks occur when adversaries employ data analysis on anonymized data to deduce sensitive information not intended for disclosure. These attacks can disclose private information about EHR owners by exploiting correlations between various data elements.Linkage Attacks: More sophisticated than simple data association, linkage attacks combine multiple pieces of information to deduce additional personal details. Attackers can re-identify individuals within a dataset by linking anonymized data with external information. Such threats are particularly relevant to blockchain-based HIE systems, where anonymized EHRs can be re-identified when combined with public datasets.Intersection Attacks: Intersection attacks involve deducing the presence of specific individuals within a dataset by intersecting multiple datasets. For instance, the intersection of known patient hospital visit times with blockchain access logs could potentially reveal patient identity.Sybil Attacks: In the context of blockchain networks, Sybil attacks involve the creation of multiple fake identities by an attacker to gain disproportionate influence within the network. This can compromise the integrity of the consensus mechanism and lead to unauthorized access or manipulation of sensitive medical data.Replay Attacks: Replay attacks capture and replay data transmissions to create unauthorized transactions or data entries on the blockchain. This can lead to the sharing of incorrect health information or compromise the integrity of healthcare information on the blockchain.Man-in-the-Middle (MitM) Attacks: Blockchain technology inherently protects data in transit through encryption and digital signatures. However, MitM attacks can occur during the block propagation process before the data is recorded on the blockchain. Attackers can intercept and potentially alter propagating blocks, leading to the dissemination of incorrect information.

The types of personal information leaks that can result from security attacks on the blockchain-based HIE system include:Identity disclosure: Identity disclosure involves leaking information that can specifically identify an individual, such as their name, date of birth, or address. Exposing identity information can be exploited for criminality, such as credit card fraud or creating fake accounts.Attribute disclosure: Attribute disclosure concerns exposing sensitive attribute information like a person’s health status, religion, ethnicity, or political beliefs. Leaking attribute information can lead to discrimination or hate crimes.Association disclosure: Association disclosure occurs when combining two or more pieces of information reveals additional private information. For example, combining a person’s location data with their social media posts could expose details about their daily life.Context disclosure: Context disclosure involves leaking the context in which personal information was collected or used. For example, disclosing a person’s search history could reveal their interests or behavior patterns.

### 3.3. Design Goal

Several requirements must be met to build a safe and efficient HIE system that resolves this trade-off problem. Because the HIE system handles EHRs, it must satisfy the security requirements to protect privacy. Moreover, it should be designed with features that facilitate efficient and meaningful EHR use to offer high data usability. Therefore, the proposed HIE system aims for the following by considering both privacy preserving and data usability:Strong Data Privacy Protection: Protecting personal information is crucial for secure EHR exchange. Patient privacy must be securely guaranteed, meaning that users participating in the blockchain network should not be able to infer the identity or information of other users from the shared data. Consequently, EHR information must be stored on the blockchain in an anonymized form using secure data protection techniques.High Data Usability: The blockchain-based HIE system requires scalability to utilize the diverse EHRs collected through a shared ledger efficiently. It should provide functionalities to utilize and process information, such as querying specific healthcare information or conducting statistical analysis from EHRs. Furthermore, it should allow independent processing of healthcare information within EHRs, enabling access to granular information. Therefore, the information in EHRs should be processed as individual transactions and stored in the blockchain.

## 4. Proposed Scheme

This section proposes a searchable blockchain-based HIE system to address the trade-off between privacy security and data usability. Users can directly utilize EHR information recorded on the blockchain without additional interactions. Moreover, users can search for specific healthcare information using search parameters to access and analyze. The healthcare information stored as anonymized transactions in the blockchain prevents privacy exposure during sharing and provides selective access control. The anonymized healthcare information offers no information to those unaware of the original data. Modifying or deleting healthcare information stored in the blockchain requires consensus from most other participants. Hence, shared healthcare information provides reliability and integrity, as it cannot be arbitrarily altered or falsified by any specific user.

### 4.1. Overall Procedure of the Searchable Blockchain-Based HIE System

The proposed searchable blockchain-based HIE system’s operation procedure is detailed in two stages: the EHR sharing and searching processes. The flow diagram of the proposed HIE system is shown in Figure 3. The process for each function is as follows.
EHR Sharing Process
Users collect healthcare information generated from various IoMT devices. Each IoMT device produces different kinds of healthcare information.Users generate EHRs using the collected healthcare information. The composition of EHRs varies based on the sensing cycle and type of IoT devices. EHRs are used to create blocks to share healthcare information on the blockchain network or to prove ownership of healthcare information stored on the blockchain.Users generate transactions by hashing the healthcare information in the EHR. The transactions consist of hashed attributes and their hashed values. Users create transactions and sign them using their secret keys. The transactions are compiled to form EHR blocks and then shared on the blockchain network.The propagated EHR blocks are connected and stored in the blockchain. Users can access and utilize the healthcare information stored in the blockchain using its search functionality.EHR Searching Process
Participants in the blockchain network use search parameters to search for healthcare information. Three types of parameters can be used: records, hashes of attributes, and hashes of values. The quantity of information provided as search results varies depending on the search parameter. Therefore, users must choose parameters appropriate to their desired level of results.The search procedure is executed based on the input search parameters. If records are input, EHR blocks with attributes and values similar to the given record are searched. The hashes of attributes and values in the record are compared with the information in EHR blocks stored in the blockchain to calculate similarity. Blocks with a similarity exceeding a user-defined threshold are returned as search results. If hashes of attributes or values are used as input, transactions matching the values are returned as search results. When searching using multiple attributes or values, similarity is calculated similarly to when using records and only results exceeding the threshold are returned.Users utilize the healthcare information from the search results according to their purposes. Blockchain-based HIE systems securely manage healthcare data, providing users with reliable information. The information can be employed for patient care, research, and policy-making, among other multifaceted applications.
Figure 3The flowchart diagram of the proposed HIE system.
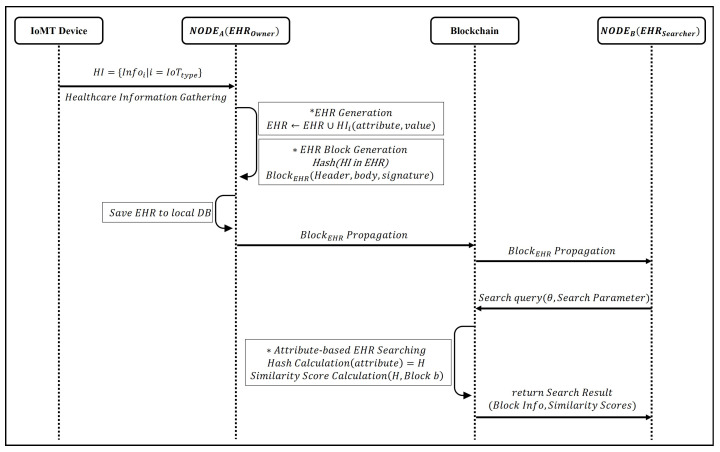


### 4.2. Initialization of the Blockchain-Based HIE System

The infrastructure for a searchable blockchain-based HIE system is set up during the initialization phase. This phase involves establishing the core components of the network and generating keys for users. Moreover, the fundamental parameters of the HIE blockchain, such as the EHR blocks’ size and generation cycle, are defined. The blockchain parameters have a direct impact on the network performance and efficiency. Therefore, the blockchain parameters in the system are defined by considering the sharing of healthcare information collected through IoT devices. As IoT devices’ sensing cycles and lifespans vary, the types and quantities of healthcare information generated have changed. Considering this, the size of EHR blocks is variably defined in alignment with the collected healthcare information and can reach a maximum of 1 MB. The block-generation cycle generates a block every time a user generates an EHR, ensuring that healthcare information remains up to date.

The generation and management of public keys (PKs) and secret keys (SKs) are crucial for robust security in the blockchain. Encryption keys are used to safeguard the identities of blockchain participants and ensure the security of transactions. The keys for the participants in the blockchain network are generated using elliptic curve cryptography (ECC). The following equations define the elliptical curves:(1)$$\begin{array}{c} y^{2}=x^{3}+ax+b\\ \mathrm{where}\;a\;\mathrm{and}\;b\;\mathrm{are}\;\mathrm{parameters}\;\mathrm{of}\;\mathrm{the}\;\mathrm{curve}. \end{array}$$
Within the finite field defined by Equation (Equation 1), a random number *n* is selected for the SK. The SK used for digital signatures and decryption must be securely managed to ensure robust security. The PK is generated through the scalar multiplication of a specific point *G* on the elliptic curve and the SK. When completing the key generation process, users share their PK with others in the network. Other participants then store these keys in the network nodes’ PK pool, essential for validating transactions in newly received blocks. We detail this transaction verification process in Section 4.4. A SK paired with a PK is managed individually by the user who created the key. A SK used for electronic signatures and decryption must be safely handled by the individual who created the key to maintain security.

New users intending to join the blockchain network must undergo user certification before generating their keys. They obtain certification from national or institutional authorities using personal identification such as passport numbers or driver’s license details. The proposed HIE system adopts a public blockchain model to increase the usability of healthcare information. Therefore, there is no central authority to control participants. Since all nodes have equal authority, we assumed a real-world authentication process for network participation to provide minimum order and security. After basic identity certification, valid users engage in the network by generating their keys following the described key generation process. New participants also receive and store the existing PK pool from adjacent nodes for future verification tasks, participating seamlessly in the network’s ongoing operations.

### 4.3. Healthcare Information Gathering

In the healthcare information gathering phase, the fundamental data for creating an EHR, denoted as healthcare information (HI), is provided. Users gather HI from various IoT devices through their smartphones or health management apps. The collected HI is utilized to monitor the user’s condition, such as providing alerts about specific health indicators or conducting trend analysis. HI is generated by measuring the user’s condition in real time or periodically using IoT devices. Depending on the type of IoT device, a wide range of information is generated, including heart rate, blood pressure, sleep patterns, and activity levels. The variety of HI collected varies depending on the sensing cycle of each IoT device. The type of information dictates the definition of HI as follows.
$$HI=\{Info_{i}|i=IoT_{type}\}$$

### 4.4. EHR and Block Generation

Users utilize the collected HI to generate EHRs and blocks. EHRs are created to integrate and manage the collected HI. The EHR generation process in the proposed system is specified in Algorithm 1.
**Algorithm 1** EHR Generation**Require:** Collected Health Information HI**Ensure:** Electronic Health Record EHR and Blockchain EHR Block $Block_{EHR}$  1:EHR←∅  2:$Block_{EHR}\leftarrow \left\{Header,Body\right\}$  3:$Header\leftarrow \{Hash_{prev},Timestamp,MerkleRoot\left(EHR\right)\}$  4:Body←∅  5:**for** 
$HI_{i}\in HI$ 
**do**  6:    $EHR\leftarrow EHR\cup \{HI_{i}\}$  7:**end for**  8:**for** 
(attribute,value)∈EHR
 **do**  9:    $attribute_{hash}\leftarrow Hash\left(attribute\right)$10:    $value_{hash}\leftarrow Hash\left(value\right)$11:    $hash_{pair}\leftarrow Hash\left(attribute_{hash},value_{hash}\right)$12:    $signature\leftarrow Sign(hash_{pair},SK_{user})$13:    $Body\leftarrow Body\cup \{\left(hash_{pair},signature\right)\}$14:**end for**15:$Block_{EHR}\leftarrow \left\{Header,Body\right\}$16:Broadcast $Block_{EHR}$ to Blockchain Network17:Store EHR in Personal Storage

Apart from personal health management, EHRs are also used to prove the ownership of EHR blocks. An EHR comprises the most recently collected HI and is defined as $EHR=HI_{1},\ldots HI_{k}$. Users generate EHR blocks as a means to share the HI contained in the EHR on the blockchain. An EHR block consists of a block header containing metadata about the block and a block body holding the data. The block header, comprising the previous block’s hash, timestamp, and Merkle root, ensures the uniqueness and integrity of the block within the network. As the proposed system does not deal with consensus algorithms, it does not address mining difficulty and nonces. The block’s body consists of transactions, which are hash values of the data to be shared. The EHR block treats the attributes and values of the HI included in the EHR as transactions. While the attributes and values of the HI exist in pairs within the transaction, they are anonymized to provide a refined search function. Attributes represent the type of HI, including various types such as age, blood sugar, location, medication information, etc. The values of HI are numerical measures of attributes sensed by IoT devices used for more practical treatment or health management. Transactions include digital signatures using the user’s SK for non-repudiation. Users who create an EHR block share it with other participants in the network. The EHR used to create the block is stored in personal storage. When participants in the network receive a newly created EHR block, they initiate a validation process as outlined in Algorithm 2. Recipients verify the block’s creation time in its header to ensure it is unique and not duplicated with any other block created simultaneously. Furthermore, they confirm that the block was created after the most recent block in the chain, ensuring its chronological validity. The next step involves validating the transactions within the block. The process includes verifying the signatures on transactions, which the EHR owner’s SK should sign, using the network nodes’ PK pool. For a transaction to be considered valid, the corresponding PK must be present in the pool, affirming the user’s legitimate participation in the network. Blocks created by a single user must not contain transactions signed by multiple users; such blocks are deemed invalid and subsequently denied. Conversely, transactions verified with a PK from the pool are accepted as valid. Lastly, the verification process includes comparing the hash value of the new block’s previous block with the last block’s hash value in the chain. This comparison ensures the integrity and continuity of the blockchain, maintaining its consistency throughout.
**Algorithm 2** Block Validation**Require:** New EHR Block, Information of Last Block, Pool of Public Keys (PK)**Ensure:** Validity of the Block  1:**Step 1:** Verify the uniqueness of the new block’s timestamp.  2:**if** timestamp of the new block exists within the same timeframe in the chain **then**  3:    Deny the new block. **return** Invalid  4:**end if**  5:**Step 2:** Check if the new block was created after the last block in the chain.  6:**if** timestamp of the new block ≤ timestamp of the last block in the chain **then**  7:    Deny the new block. **return** Invalid  8:**end if**  9:**Step 3:** Validate transactions within the block.10:**for** each transaction in the new block **do**11:    **Step 3.1:** Verify the transaction signature using the PK pool.12:    **if** signature of the transaction cannot be verified with any PK in the pool **then**13:        Deny the new block. **return** Invalid14:    **end if**15:    **Step 3.2:** Ensure the transaction is signed by a single user.16:    **if** transaction contains signatures from multiple users **then**17:        Discard the new block. **return** Invalid18:    **end if**19:**end for**20:**Step 4:** Verify the previous block’s hash value.21:**if** previous hash value in the new block ≠ hash value of the last block in the chain **then**22:    Discard the new block. **return** Invalid23:**end if**24:Add the new block to the chain. **return** Valid

### 4.5. EHR Searching Algorithm

In the proposed system, users can obtain the desired HI through an attribute-based search algorithm. Parameters for the search may include the hash of an EHR, the hash of a specific attribute header, or the hash of an attribute value. Depending on the type of parameter used in the search, users can employ data similarity. By adjusting the threshold during data matching, users can either obtain specific information as search results or choose to receive broader statistical information. The attribute-based healthcare information search algorithm in the proposed system is specified in Algorithm 3. Algorithm 3 initiates by preparing the blockchain database for a comprehensive search. Users input various search parameters such as an EHR’s hash, specific attribute hashes, or hashes of attribute values. For each parameter, the algorithm determines its nature. If the parameter is an EHR, the HashCalculation function (Algorithm 4) is called to hash each attribute using the SHA-256 algorithm, ensuring a secure format for comparison with blockchain data. In cases where the parameter is a single attribute or a set of hashed values, the algorithm processes these inputs by hashing the single attribute or utilizing the provided hashes. This standardization of input data into a hash format aligns with the blockchain’s structure.

In the search process of our proposed HIE system, the calculation of similarity scores between a search query and the data stored on the blockchain is pivotal in ensuring the relevance and precision of the search outcomes. The SimilarityScoreCalculation function, delineated in Algorithm 5, is instrumental in computing these similarity scores for each block within the blockchain relative to the search query. The process commences with the identification of the number of hashes from the search query (*H*) that do not find a corresponding match within a specific blockchain block (*b*), referred to as mismatchCount. The similarity score is then determined using the formula $1-\frac{\mathrm{mismatchCount}}{\mathrm{size}\;\mathrm{of}\;H}$, where the size of *H* denotes the total number of hashes in the search query. This formula inversely relates the proportion of mismatched hashes to the total query hashes to calculate the similarity score. Thus, a lower mismatchCount yields a similarity score approaching 1, indicating a higher likeness between the search query and the blockchain block. In contrast, an increased count of mismatches results in a score diminishing to 0, suggesting a lesser degree of similarity.
**Algorithm 3** Attribute-based EHR Searching Algorithm**Require:** Search Threshold θ, Search Parameters**Ensure:** Search Results (Blocks’ Information and Similarity Scores)  1:Initialize Blockchain Database  2:**for** each Search Parameter *p* **do**  3:    Let *H* be the set of hashes for parameter *p*  4:    Let *S* be an empty set for storing similarity scores  5:    **if** *p* is an EHR **then**  6:        H← HashCalcaulation(EHR’s attributes)  7:    **else if** *p* is an Attribute **then**  8:        H←{Hash(p)}  9:    **else if** *p* is a set of Hashed Attribute Values **then**10:        H←p11:    **end if**12:    **for** each block *b* in Blockchain **do**13:        score←SimilarityScoreCalculation(*H*, *b*)14:        Add (b,score) to *S*15:    **end for**16:    Filter *S* to include only those with score >θ17:**end for**18:**return** 
*S*

**Algorithm 4** Hash Calculation
1:**function** HashCalculation(Attributes)2:    Initialize an empty set Hashes3:    **for** each attribute attr in Attributes **do**4:        Compute hash← SHA256(attr)5:        Add hash to Hashes6:    **end for**7:    **return** Hashes8:
**end function**



**Algorithm 5** Similarity Score Calculation
1:**function** SimilarityScoreCalculation(Hashes *H*, Block *b*)2:    Let mismatchCount←03:    **for** each hash *h* in *H* **do**4:        **if** *h* does not match any hash in block *b* **then**5:           mismatchCount←mismatchCount+16:        **end if**7:    **end for**8:    **return** $1-\frac{mismatchCount}{\mathrm{size}\;\mathrm{of}\;H}$9:
**end function**



Consider a scenario where a search query consists of five hashes (H=5) for a more concrete understanding. Suppose a specific blockchain block and, out of the five hashes from the search query, three hashes match those within the block, resulting in two hashes that do not match (mismatchCount=2). According to our similarity score formula $1-\frac{\mathrm{mismatchCount}}{\mathrm{size}\;\mathrm{of}\;H}$, the similarity score for this blockchain block would be calculated as $1-\frac{2}{5}=0.6$. Only those blocks that achieve a similarity score exceeding a pre-established threshold θ are deemed pertinent and thus selected for inclusion in the search results. This example demonstrates how the similarity score effectively filters the blockchain blocks to include only those most relevant to the user’s search query, enhancing the precision of the search within the blockchain-based HIE system.

### 4.6. Search Result

In the proposed system, an EHR searcher receives information about blocks and transactions due to their search. EHR searchers who recognize the search results’ original meaning can effectively utilize the hashed HI without a separate decryption process. This feature enables them to access information about the distribution or frequency of specific HI within the blockchain network. To further assist the analysis, Figure 4 and Figure 5 provide visual representations of the searching process and outcomes for particular HI. These graphs deliver insightful information, such as the number of records with matching attributes or the specific values of those attributes. Blockchain participants can directly use the HI stored in the system or leverage the aggregated statistical data for various applications, including tracking infectious diseases, conducting medical AI research, or implementing personalized healthcare management strategies. Importantly, the proposed system ensures data privacy by employing hashed data during the EHR sharing and searching processes, thus safeguarding against security threats.

## 5. Experiment and Results

### 5.1. Experimental Setup

To verify whether the proposed HIE system mitigates the existing trade-off problem and enables safe and convenient sharing of healthcare information, a simulation was designed. The simulation environment is based on the EHR sharing process defined in [28]. The scenario assumes 1000 users in a blockchain network sharing anonymized EHRs in a random sequence. The EHRs used in the simulation were synthetic EHRs provided by Synthea. The HIE blockchain was implemented using Hyperledger 2.3.1 [29] and Apache CouchDB [30] was used as the state database. For the simulation, a PC equipped with 32 GB memory and a 3.80 GHz Intel Core i7-10700K central processing unit was utilized in a wired environment. This study does not cover improvements to the consensus process performed on the blockchain network. Therefore, these improvements were not evaluated.

### 5.2. Synthetic EHR

The simulation uses patient-related data from a set of synthetic EHRs of COVID-19 patients provided by *Synthea* (https://synthea.mitre.org/downloads, accessed on 12 July 2023). Table 1 describes 25 attributes in the synthetic data set used for the simulation. Attributes in the synthetic records are broadly classified into five categories: identifiable information about the patient, personal information, demographic information, geographical information, and detailed health-related information. The level of identifiability of a subject varies depending on the type of attribute, especially when exposed to others. Therefore, caution is needed to minimize personally identifiable information in attributes when sharing EHRs. A total of 12,352 synthetic records were used in the simulation. In the proposed blockchain-based HIE system, each record is processed as one EHR block. Hence, including the genesis block, the simulation was conducted using a blockchain storage comprising a total of 12,353 blocks.

### 5.3. Performance Analysis

In this section, the search performance of the proposed system is evaluated by measuring the search time and the number of records found, based on the number of attributes used in the search. The primary metric for performance evaluation, elapsed time, is measured in terms of average, shortest, and longest durations. For each phase, a search simulation was conducted using randomly selected attributes from the 25 attributes of a specific record. The results of the healthcare information search simulation in the blockchain, based on the number of attributes used, are shown in Table 2.

It was shown that the search time increases linearly as the number of attributes used in the search increases. Searches using fewer attributes show faster times due to fewer information comparisons. However, the number of attributes is not the only factor affecting search performance. If the attributes used are unique, it becomes difficult to find matching properties. In cases where a unique attribute with a low matching probability was chosen in searches using one attribute, it was found to take 2632.05 ms to complete the search. Conversely, in searches using multiple attributes, if attributes with a high probability of matching are used, the search can be completed in a shorter time. Therefore, it was confirmed that the number and type of attributes used in the proposed attribute-based search algorithm influence the results.

As the number of records used increased, it was observed that more similar records were found. However, beyond a certain number of attributes, the number of similar records found began to decrease. The highest number of records, 3974, was found using 15 attributes. From the results of searches using more than 16 attributes, a decrease in the number of records found was observed. As the number of attributes used in the search increases, the probability of finding similar records increases due to the ability to compare various items. However, if the number of attributes is too many or too few, only accurate matching results are handled, reducing the probability of finding similar records. The simulation results confirmed that the proposed searchable HIE system can provide optimal search performance depending on the number and type of attributes used by the user. Poor search performance when users use inaccurate search parameters can be interpreted. However, this implies strong resilience against random information collection attacks using unspecified parameters in the blockchain-based HIE system, typically employed to protect user privacy. Therefore, the proposed system demonstrates its effectiveness in resolving the trade-off problem in HIE by providing strong privacy protection performance and high usability.

### 5.4. EHR Sharing Simulation

We conducted a series of experiments to assess the performance of the proposed HIE system, with a specific focus on the delay encountered during the HI sharing process. A blockchain network simulation involving 1000 nodes was implemented to measure the delay. This simulation measured the duration from when an EHR block is generated by a randomly selected node to its dissemination among other nodes. It aimed to reproduce the continuous collection and sharing of varied HI by IoMT devices in real-world scenarios. Synthetic health datasets provided by Synthea were utilized to simulate the HI sharing process.

The outcomes of the block propagation experiment within our HIE system are depicted in Figure 6. Using the gossip protocol for data propagation, we found that the average propagation time between nodes was 5.46 milliseconds. Notably, certain instances exhibited pronounced delays in the node-to-node information transfer, with outlier values averaging 70.56 milliseconds. The outlier values suggest that, in worst-case scenarios, where a node disseminates a block to numerous others, the propagation process via the gossip protocol can become significantly time-consuming.

Figure 7 illustrates the total execution time required to share EHR information through the blockchain, capturing the entire process from block creation using collected HI to its network-wide propagation. The average total time was recorded at 26.58 s. These results affirm that the proposed HIE system possesses adequate performance capabilities to support a broad spectrum of patient condition monitoring services, ensuring timely information delivery even in emergency scenarios where swift patient care is crucial. Through these experimental findings, we demonstrate the efficacy of our HIE system in facilitating real-time HI sharing via blockchain.

### 5.5. Security Performance Simulation

We performed a series of simulations to evaluate the data confidentiality of the proposed HIE system. The synthetic health dataset from Synthea employed in our simulations encompasses a diverse array of HI attributes. The uniqueness of an attribute within the dataset is determined by the frequency of its values’ occurrence. The dataset consisted of 25 attributes, with ten identified as unique attributes. More unique attributes contribute to a higher difficulty in re-identification, thereby enhancing resistance to re-identification attacks. Details regarding these attributes are provided in Table 3.

Given the variability of HI types collected via the IoMT devices, the composition of EHRs generated from this collected HI varies with each collection. We evaluate the data confidentiality of the proposed HIE system against security threats based on including properties unique to the EHR. To this end, we performed simulations ranging from an EHR with no unique attributes to one with unique attributes. Each simulation scenario was executed 1000 times, varying the number of attributes. Figure 8 shows the results of the re-identification attack depending on the number of unique attributes that make up the EHR. We observed that the success rate of re-identification attacks on EHRs constituted solely of normal attributes averaged 21.76%, while the rate for EHRs composed entirely of unique attributes averaged close to 0%. As all HI within the proposed system is shared anonymously using SHA-256 encryption, re-identification attacks are inherently challenging, providing a baseline resistance. However, the simulations revealed vulnerabilities to re-identification attacks based on the types of attributes composing the EHRs. The results of the attribute-based processing of EHRs demonstrate that the security robustness varies with the types of HI comprising the EHRs. Unique attributes, by their nature, are data types that are unlikely to match, thus, if disclosed, they could be exploited to identify individuals. Yet, the difficulty in re-identifying and extracting information through unique attributes renders the risk of privacy exposure exceedingly slim. In conclusion, the simulation results substantiate our proposition of an HIE system capable of securely sharing HI against security threats.

### 5.6. Security Analysis

This section delves into a detailed analysis of the security threats discussed in the Section 3.2, evaluating their potential impact on the blockchain-based HIE system and discussing countermeasures to mitigate these risks.

Inference Attacks: Blockchain technology’s transparency allows all network participants to access the stored data, posing a risk of inference attacks by malicious nodes or users with honest curiosity. In such attacks, sensitive details can be deduced from anonymized data, breaching the confidentiality of EHR. Our proposed HIE system counters this by processing EHR data attribute-wise, treating each as a separate transaction. This method minimizes the amount of HI that can be leakage. Consequently, even in a breach, the exposed HI is fragmented, offering attackers only partial information about the EHR. This strategy makes it difficult for malicious users to identify the EHR owner through inferred information or to use it for criminal purposes. The proposed system does not employ advanced encryption techniques requiring high computational complexity to improve data usability. It is also based on a public blockchain model to increase accessibility to HI. Therefore, security may be relatively weak compared to an HIE system based on a private blockchain model using advanced encryption technology such as homomorphic encryption. To supplement the security of the proposed system, you can consider applying data masking technology to the HI information to be shared or using categorized values. Alternatively, you can consider applying access control technology through a private blockchain so that only verified users can participate in the network.Linkage Attacks: Linkage attacks threaten patient anonymity by potentially re-identifying individuals through correlating anonymized HI with external data sources, risking unauthorized privacy disclosure. The proposed system refrains from storing combined or linked data that may expose personal information to prevent linkage attacks. For example, it does not aggregate location data and social media posts with blockchain-stored information, treating each data piece separately and securely to avoid unintended comprehensive disclosures about an individual’s daily activities. In combating linkage attacks, the proposed HIE system employs SHA-256, a robust anonymization technique, to hash all HI shared within the blockchain network. This approach effectively deals with identifiable features in the original data, ensuring anonymized HI does not contain direct identifiers. The unidirectional nature of SHA-256 makes it virtually impossible for attackers to reconstruct the original data from the hash value, significantly hindering their ability to infer the original HI. SHA-256’s high collision resistance and sensitivity mean even minor changes in input data cause drastic alterations in the hash output, further deterring the re-creation of original data. Consequently, applying SHA-256 for anonymization of HI in the proposed HIE system enhances the reliability and integrity of the shared data and provides a solid foundation for safeguarding user privacy.Intersection Attacks: Intersection attacks exploit the overlapping information among different datasets to identify individuals within the blockchain network, breaching user anonymity and privacy. Adopting privacy-preserving data aggregation methods, like secure multi-party computation (SMPC), is recommended to mitigate the risks associated with intersection attacks. We propose an HIE system that leverages blockchain technology to preserve user privacy. Before storing data, this system applies a robust anonymization or pseudonymization process using cryptographic algorithms to strip or alter personally identifiable information. This approach maintains the data’s integrity while concealing any direct identifiers. Additionally, the system employs an attribute-based approach to handle HI, fragmenting sensitive EHR information into multiple parts. Therefore, even if some HI is exposed to an attacker, it is impossible to specify the actual identity of the EHR owner because the attacker cannot reconstruct all the EHR information.Sybil Attacks: In the proposed HIE system, participants in the blockchain network directly create and propagate EHR blocks. As such, the system does not utilize consensus mechanisms like Proof of Work (PoW) or Proof of Stake (PoS) but instead employs a Practical Byzantine Fault Tolerance (PBFT)-based process to maintain the integrity of the blockchain ledger. The proposed HIE system adopts a method to limit users’ issuance and use of multiple keys as a countermeasure against Sybil attacks. It is assumed that users must authenticate their identity using their passport number or driver’s license information and a unique identifier from a country or authority to participate in the network and generate keys. Each participant, having undergone verification, is allocated a single SK, ensuring they possess only one vote in any PBFT consensus activity. This limitation prevents users from gaining undue influence over the ledger, protecting the network from unauthorized alterations or deletions.Replay Attacks and Man-in-the-Middle (MitM) Attacks: During the propagation of blocks within the blockchain network, attackers could intercept blocks, alter their data, or substitute them with incorrect information, thereby compromising the integrity and consistency of health data within the HIE system. Implementing nonce and timestamp mechanisms in blockchain transactions is essential to prevent such attacks. In the proposed HIE system, users generate blocks using their EHR data, eliminating the need for a consensus process for block creation. As a result, there is no nonce value associated with block creation. Instead, each EHR block records the creation time in its block header during the generation process. Furthermore, all transactions within the block body are digitally signed with the private key of the EHR owner who generated the block. This signature safeguards against attackers intercepting the block during propagation, modifying its content, or regenerating blocks with incorrect information. Before appending a block to the chain, participants verify the block’s information and the transaction signatures they receive, ensuring the integrity and authenticity of the data. These mechanisms provide the uniqueness of each transaction and confine its validity to a specific timeframe, safeguarding the system against unauthorized or duplicate transactions.

## 6. Conclusions

This study proposes a searchable blockchain-based HIE system to address the trade-off between security and data usability. The proposed system utilizes an attribute-based granular approach for EHR processing. Therefore, it enables the efficient search and utilization of EHRs without unnecessary decryption processes. The proposed system allows users to perform selective queries and statistical analyses on specific healthcare information, offering enhanced data usability. Our systems employ blockchain technology’s immutability and decentralization attributes to safeguard EHRs against the potential privacy threats inherent in HIE systems. All healthcare information is anonymously recorded in the ledger through hash algorithms, preventing attempts to deduce user identities from the disclosed data. Modifications to or deletions of data within the HIE blockchain require the consensus of most network participants. This consensus mechanism ensures that, once data is stored on the blockchain, altering it without network participants’ agreement becomes virtually unfeasible, guaranteeing the stored healthcare information’s integrity and trustworthiness. Performance and security analyses confirmed that the proposed system enables accurate and detailed searches using data similarity and attributes. Simulation results indicated that searches using 15 out of 25 attributes yielded the highest number of records, averaging 3974. These results demonstrate that effective searches can be conducted using only a subset of the total record information. Consequently, this aligns with our design objective to enhance system efficiency while minimizing privacy leakage threats by reducing information sharing.

While the proposed blockchain-based HIE system has demonstrated significant advancements in balancing security with data usability, certain limitations warrant discussion. One notable limitation is the scalability problem as the volume of EHRs grows exponentially. While beneficial for security, the blockchain’s immutable nature poses challenges regarding storage and processing efficiency, potentially leading to increased transaction times and costs. Moreover, while the attribute-based granular approach enhances data usability, it also introduces complexity in managing access controls and ensuring that fine-grained permissions are accurately enforced. This complexity may impact the system’s adaptability to diverse healthcare settings with varying regulatory and operational requirements. Future work will focus on optimizing the blockchain infrastructure to improve scalability and reduce transaction overhead. Additionally, we consider enhancing the security of HIE systems through research on the lightweight implementation of advanced cryptographic techniques such as homomorphic encryption and searchable encryption. By addressing these challenges, the proposed system will aim to provide a more robust solution that can adapt to the dynamic needs of the healthcare industry, ultimately leading to wider adoption and more effective healthcare information exchange. 

## Figures and Tables

**Figure 1 sensors-24-01582-f001:**
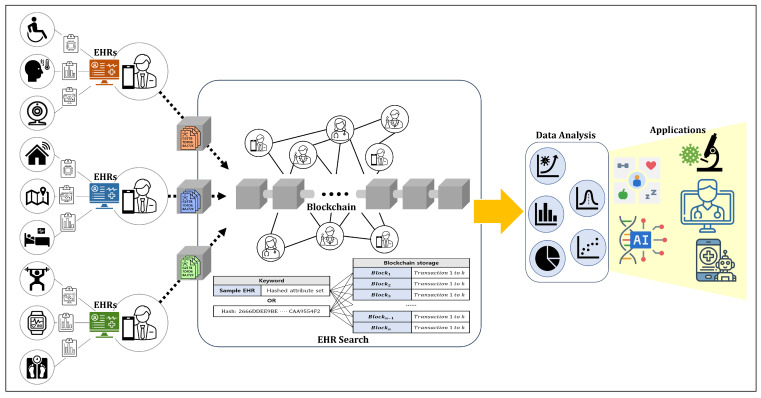
Overview of the proposed searchable blockchain-based HIE system.

**Figure 2 sensors-24-01582-f002:**
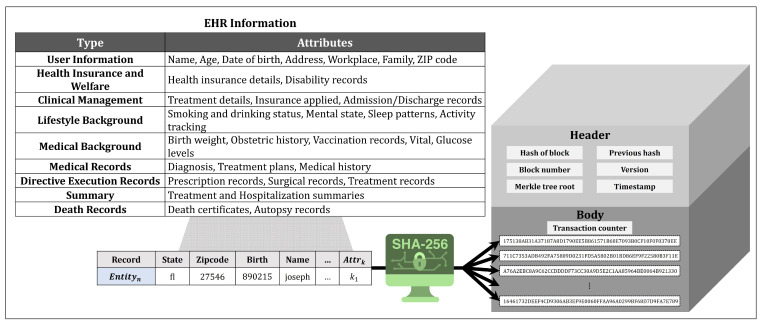
Proposed block structure of the blockchain-based HIE system.

**Figure 4 sensors-24-01582-f004:**
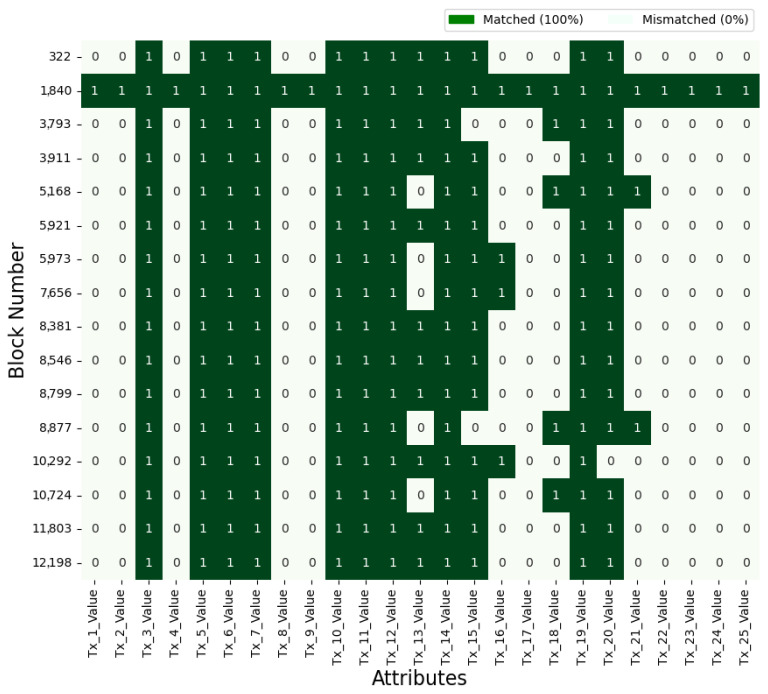
Healthcare information search results based on attributes.

**Figure 5 sensors-24-01582-f005:**
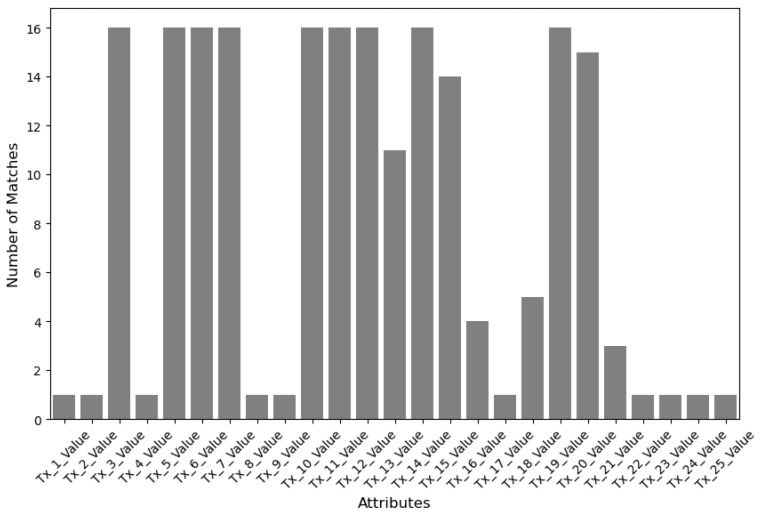
Frequency of matching attributes.

**Figure 6 sensors-24-01582-f006:**
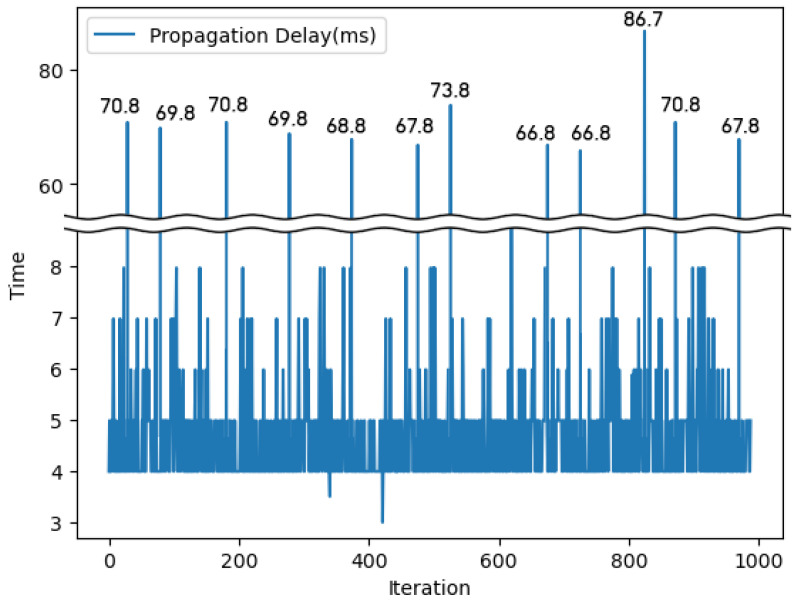
Propagation time of blocks between nodes.

**Figure 7 sensors-24-01582-f007:**
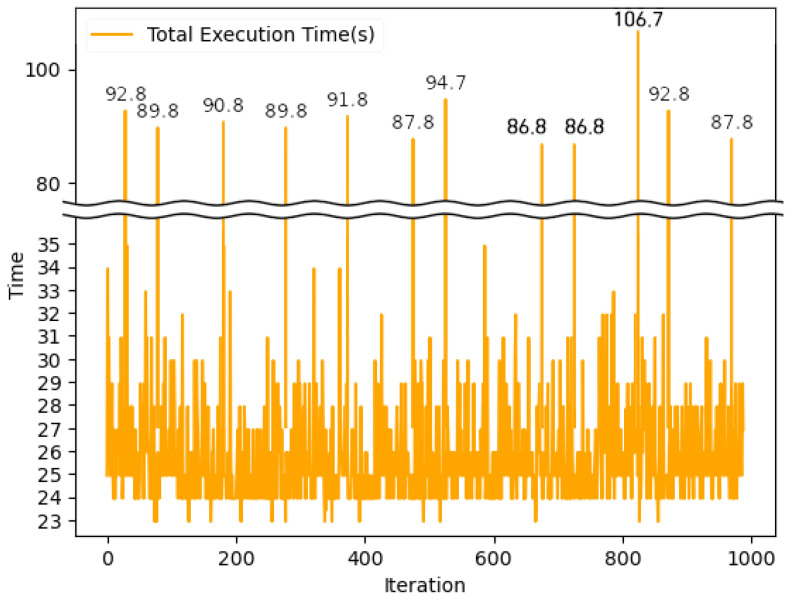
Total execution time for sharing EHR information.

**Figure 8 sensors-24-01582-f008:**
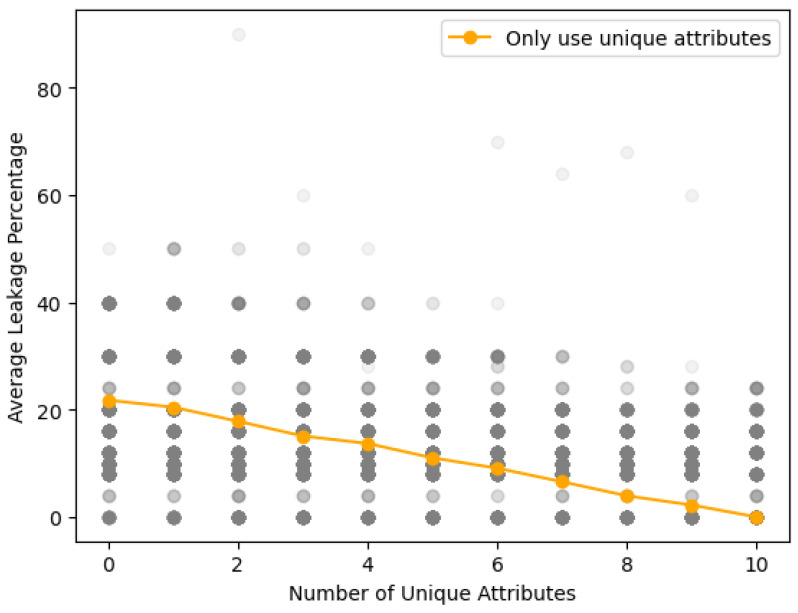
Re-identification attack success rate according to the number of unique attributes.

**Table 1 sensors-24-01582-t001:** Attributes of the COVID-19 patient dataset provided by *Synthea* [31].

Category	Fields	Description
Identification	ID	Unique identifiers for patients.
	SSN	Social Security Numbers.
	Drivers	Driver’s license numbers.
	Passport	Passport numbers.
Personal Information	Birthdate	The birth dates of patients.
	Deathdate	Dates of death, if applicable.
	Prefix, First, Last, Suffix	Parts of the patient’s name.
	Maiden	Maiden names, if applicable.
	Marital	Marital status.
Demographics	Race, Ethnicity, Gender, Birthplace	Demographic details.
Address	Address, City, State, County, ZIP	Address details.
	Latitude, Longitude	Geographical coordinates.
Healthcare Details	Healthcare Expenses and Coverage	Financial details related to healthcare.

**Table 2 sensors-24-01582-t002:** Healthcare information searching time based on the number of attributes.

Number of Attributes	Shortest Elapsed Time (ms)	Longest Elapsed Time (ms)	Average Elapsed Time (ms)	Average Found Records
1	8.09 ms	2632.05 ms	98.65 ms	1
2	8.21 ms	2693.75 ms	105.96 ms	2.4
3	9.19 ms	1957.81 ms	115.41 ms	1.26
4	10.52 ms	2200.12 ms	123.65 ms	1.27
5	12.46 ms	2018.36 ms	140.73 ms	266.41
6	13 ms	3101.46 ms	145.6 ms	266.13
7	13.5 ms	2623.53 ms	157.53 ms	402.94
8	15.53 ms	1919.72 ms	168.18 ms	266.16
9	16.76 ms	2475.46 ms	180.43 ms	266.44
10	17.96 ms	2567.42 ms	187.7 ms	407.75
11	18.57 ms	1862.05 ms	200.53 ms	406.39
12	20.27 ms	2702.53 ms	217.24 ms	1151.62
13	20.84 ms	2960.34 ms	224.84 ms	925.76
14	22.15 ms	2311.72 ms	235.41 ms	2380.34
15	23.4 ms	3361.02 ms	245.36 ms	3974.07
16	24.07 ms	2362.89 ms	248.45 ms	2218.18
17	24.96 ms	3082.44 ms	266.1 ms	2218.18
18	25.38 ms	2451.37 ms	266.68 ms	801.86
19	26.35 ms	2559.06 ms	275.97 ms	2247.65
20	27.16 ms	2994.91 ms	285.51 ms	2483.63
21	28.03 ms	3036.52 ms	295.47 ms	1308.84
22	28.51 ms	2482.71 ms	299 ms	1308.84
23	30.61 ms	2290.12 ms	311.63 ms	484.46
24	30.99 ms	3518.19 ms	324.51 ms	484.47
25	32.53 ms	2759.03 ms	331.23 ms	485.78

**Table 3 sensors-24-01582-t003:** The type of attribute.

Unique Attributes	Normal Attributes
‘DRIVERS’, ‘ADDRESS’, ‘PASSPORT’, ‘LAT’, ‘LON’, ‘MAIDEN’, ‘SSN’, ‘HEALTHCARE COVERAGE’, ‘ID’, ‘HEALTHCARE EXPENSES’	‘MARITAL’, ‘COUNTY’, ‘CITY’, ‘BIRTHDATE’, ‘STATE’, ‘LAST’, ‘ETHNICITY’, ‘FIRST’, ‘RACE’, ‘BIRTHPLACE’, ‘ZIP’, ‘GENDER’, ‘PREFIX’, ‘SUFFIX’, ‘DEATHDATE’

## Data Availability

Synthea: https://synthea.mitre.org/downloads (accessed on 12 July 2023).

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
