# Peer review of "Searchable Blockchain-Based Healthcare Information Exchange System to Enhance Privacy Preserving and Data Usability"

_sensors, 2024, doi:10.3390/s24051582_

Round 1

Reviewer 1 Report

Comments and Suggestions for Authors

1. Expand the threat model section to include more potential privacy attacks and detail how the system guards against them.

2. Provide pseudocode or algorithms for the core functions like EHR creation, blockchain validation, and similarity score calculation.

3. Include performance metrics like throughput in transactions per second or latency during evaluation.

4. Clarify the process for adding new participants and distributing keys in the initialization phase.

5. Explain the approach used for validating identity and preventing sybil attacks.

6. Do a few direct studies from application perspective; cite and reference them to strengthen the paper:

a) Thantharate, P.; Thantharate, A. ZeroTrustBlock: Enhancing Security, Privacy, and Interoperability of Sensitive Data through ZeroTrust Permissioned Blockchain. Big Data Cogn. Comput. 2023, 7, 165. https://doi.org/10.3390/bdcc7040165

b) Ali, A.; Al-rimy, B.A.S.; Alsubaei, F.S.; Almazroi, A.A.; Almazroi, A.A. HealthLock: Blockchain-Based Privacy Preservation Using Homomorphic Encryption in Internet of Things Healthcare Applications. Sensors 2023, 23, 6762. https://doi.org/10.3390/s23156762

c) Hussien, H.M.; Yasin, S.M.; Udzir, N.I.; Ninggal, M.I.H. Blockchain-Based Access Control Scheme for Secure Shared Personal Health Records over Decentralised Storage. Sensors 2021, 21, 2462. https://doi.org/10.3390/s21072462 

Author Response

Dear Reviewers,

Most of all, we would like to thank the associate editor and the reviewers for spending their valuable time and effort reviewing our paper. We have carefully read all of the reviewers' comments and have revised our manuscript according to reviewers' insightful comments and suggestions. Please find below our detailed replies to each of the comments.

Once again, we appreciate your kind and careful suggestions.

Sincerely,

Sunghyun Cho, Prof./Ph.D.

Dept. of Computer Science and Engineering.

Hanyang University ERICA

55 Hanyangdaehak-ro, Sangrok-gu, Ansan, Gyeonggi-do, Korea.

Tel: +82-31-400-5670 / Fax: +82-31-436-8152

Reviewer 2 Report

Comments and Suggestions for Authors

This paper proposes a blockchain-based healthcare information exchange (HIE) system that aims to balance privacy protection and data usability. The system allows searching for specific health data on the blockchain using hashed attributes. The authors claim this approach enhances utility without compromising security. The paper is generally well-written, and the motivation for the work is strong. However, I do have some comments and suggestions that I hope will strengthen the paper:

1. The threat model presented in Section 3.2 is a bit thin. I think the authors could expand on the types of attacks that are possible in this system, especially given the sensitivity of health data. More details on the capabilities of attackers would help motivate the design choices.

2. In Section 4.5 on the searching algorithm, it is not completely clear to me how "similarity" scores are calculated between a search query and the blockchain data. Some more details on the matching criteria would be helpful. Perhaps include a small example.

3. The security analysis in Section 5.4 is also quite brief. The authors claim the system protects against various types of disclosure, but do not provide evidence for these claims. More analysis on exactly how the system defends against specific attack vectors would strengthen this section.

4. The experimental evaluation focuses heavily on performance, but does not evaluate security and privacy aspects. Supplementary experiments showing how the system protects data confidentiality would be beneficial.

5. In Section 3.1.4, "EHR searcher" is introduced but not fully defined until Section 4.5. Consider briefly describing their role earlier.

6. Table 2 could be improved by including units on the elapsed time columns.

7. Consider more figures to illustrate the system architecture and workflow. This would provide helpful visual summaries.

8. The conclusion is quite short. Expanding on the limitations and future work would provide better closure.

As a conclusion, I found this paper makes a nice contribution in an important area. With additional revision to address the comments, I believe the work would be suitable for publication. 

Comments on the Quality of English Language

* The paper is generally well-written, but there are some minor grammatical and word choice issues that should be addressed.

* Be consistent with verb tenses. Some sentences shift between past and present tense. Using present tense to describe the proposed system would be preferable.

* Avoid unnecessary words like "of the" and "for the." For example, in the abstract "of the proposed system" can just be "proposed system."

* Some sentences are overly long or complex. Breaking them into shorter, simpler sentences could improve readability.

* There are some typos to fix such as missing words or punctuation. Carefully proofread the paper.

* Some terms are overused when other wording would be better. For example, use "method" or "approach" rather than "technology" in some cases.

* The tone is appropriately technical for a research paper, but avoid sounding too colloquial in places.

Author Response

(The authors gave the same response as above.)
